# Active Compliance Smart Control Strategy of Hybrid Mechanism for Bonnet Polishing

**DOI:** 10.3390/s24020421

**Published:** 2024-01-10

**Authors:** Ze Li, Chi Fai Cheung, Kin Man Lam, Daniel Pak Kong Lun

**Affiliations:** 1State Key Laboratory of Ultra-Precision Machining Technology, Department of Industrial and Systems Engineering, The Hong Kong Polytechnic University, Hong Kong, China; ze036.li@connect.polyu.hk; 2Department of Electrical and Electronic Engineering, The Hong Kong Polytechnic University, Hong Kong, China; kin.man.lam@polyu.edu.hk

**Keywords:** active compliance control, self-optimisation control system, hybrid mechanism, force control, robotic polishing

## Abstract

Compliance control strategies have been utilised for the ultraprecision polishing process for many years. Most researchers execute active compliance control strategies by employing impedance control law on a robot development platform. However, these methods are limited by the load capacity, positioning accuracy, and repeatability of polishing mechanisms. Moreover, a sophisticated actuator mounted at the end of the end-effector of robots is difficult to maintain in the polishing scenario. In contrast, a hybrid mechanism for polishing that possesses the advantages of serial and parallel mechanisms can mitigate the above problems, especially when an active compliance control strategy is employed. In this research, a high-frequency-impedance robust force control strategy is proposed. It outputs a position adjustment value directly according to a contact pressure adjustment value. An open architecture control system with customised software is developed to respond to external interrupts during the polishing procedure, implementing the active compliance control strategy on a hybrid mechanism. Through this method, the hybrid mechanism can adapt to the external environment with a given contact pressure automatically instead of relying on estimating the environment stiffness. Experimental results show that the proposed strategy adapts the unknown freeform surface without overshooting and improves the surface quality. The average surface roughness value decreases from 0.057 um to 0.027 um.

## 1. Introduction

Ultra-precision polishing is a process that aims to achieve nanometre-level surface quality and micrometre-level form accuracy on various materials [1,2]. It is one of the most important technologies in some high-end application scenarios, such as the optical industry, biomedical industry, and aerospace applications [3]. In recent decades, with the wide utilisation of these high-precision complex freeform surface components, researchers have tried to introduce some advanced techniques in ultraprecision finishing processes to enhance the performance of finishing [4,5,6]. To fulfil the increasing demands for high-quality products, compliance control strategies and advanced computer-controlled ultra-precision polishing processes have been employed, such as small tool polishing (STP) [7], magnetorheological finishing (MRF) [8], plasma polishing [9,10], stressed-lap polishing (SLP) [11], fluid jet polishing (FJP) [12,13], and bonnet polishing (BP) [14,15]. Compliance control has attracted much attention in recent years [16]. In Zhu’s review, the compliance strategies for polishing and grinding can be divided into five levels: holder compliance, tool compliance, slurry compliance, abrasive compliance, and workpiece compliance [17]. Among these compliance strategies, we only focus on mechanical-related techniques, tool compliance, holder compliance, and some workpiece compliance methods. Most mechanical-related compliance strategies are based on polishing mechanisms such as robotics, workpiece holders, and polishing tools with complex designs and structures [18,19,20,21]. Nevertheless, there is little research on the compliance control strategy of the main mechanism of the machine. Although tool compliance, holder compliance, and workpiece compliance can improve the performance of the polishing process, compliance with the machine’s main mechanism is still a non-negligible part.

As one of the ultra-precision polishing processes, bonnet polishing is a remarkable process with excellent machining performance [22,23]. It applies the unique precession movement and a rotating inflated spherical tool [24]. At the end of the 1990s, the Zeeko Company and David Walker from the University College London first applied the BP technique to polish optical components, especially for components with aspherical and freeform surfaces. The Zeeko IRP series is shown in Figure 1 [14,15,24,25,26]. Figure 2 shows some geometric parameters for the bonnet polishing process, such as angular velocity ω, inclination angle φ, contact spots, etc. The unique precessions motion eliminates the zero-velocity point from the contact spot with a near-Gaussian material removal function [24].

In the case of the control system of the bonnet polishing machine, multi-axis computer-controlled ultra-precision polishing (CCUP) technology is utilised to fabricate ultra-precision freeform surfaces with sub-micrometre form accuracy and nanometre surface roughness [1,28]. Usually, a BP polishing machine comprises two parts: a three-axis movement platform and a rotatory part consisting of a spindle with a polishing head.

Previously, our team developed a hybrid mechanism for the bonnet polishing process with the advantages of the serial mechanism and parallel mechanism, which can be used to solve the drawbacks of classical polishing machine tools and industrial robotic polishing systems, such as insufficient stability, limited positioning accuracy, low stiffness, and unexpected vibration, achieving stable removal effectiveness in a sub-aperture region [29]. Due to the characteristics of the parallel mechanism, such as higher precision, stiffness, and load-carrying capacity, this hybrid polishing mechanism can be used for high-precision application scenarios [30]. For instance, because of the high-precision characteristic inherited from the parallel mechanism, the resolution of the incremental position of the end-effector can achieve 1 um without calibration in this work, which is fundamental to the position-based force control strategy. The prototype of the hybrid polishing mechanism is illustrated in Figure 3 [29].

However, the current mainstream concept of the precision polishing procedure still follows the classical steps, which are based on the pre-measurement of the workpiece to generate the corresponding polishing toolpath and output the predefined trajectory to the control system of the polishing machine. This method is time-consuming, ineffective, and strongly depends on the accuracy of the measurement and polishing machine and toolpath generation algorithms, limiting the performance of the polishing machine. A crucial drawback is that this method is an open-loop method without timely online correction to compensate for the uncertainty during the polishing process. Specifically, during the polishing process, a polishing head is required to interact with the external environment, the sensitive workpiece, and with constrained contact pressure [23,31]. Especially in the ultraprecision manufacturing process, the controlled contact pressure can provide ideal situations to enhance the machine’s performance [32]. With the controlled and suitable contact pressure applied by the tool on the workpiece, the polishing process can ensure a consistent material removal rate, avoid excessive wear or damage, and adapt to the surface geometry and hardness variations [33,34,35]. Nevertheless, most commercial controllers are integrated and sealed, especially the commercial polishing machine, which only allows researchers to generate accurate polishing toolpaths based on time-consuming pre-measurements and implementing generated G-code line by line. This direct G-code implementation method ignores the position errors from the motion system of the machine at a real-time level. Also, users can only monitor the process parametre without the authority to correct it online [29]. As a result, they cannot exploit advanced control strategies to optimise the polishing process by compensating for the polishing trajectory or parametres online. Therefore, establishing an open architecture and scalable polishing control system with a customised force controller for the ultra-precision polishing process is necessary to extend and explore the new techniques of the ultra-precision polishing process.

On the other hand, compliance control strategies can be classified into two categories: passive compliance and active compliance [36,37]. In this research area, most compliance control strategies are based on passive compliance [17,31,38], which may lead to insufficient stiffness and cause a lower polishing efficiency. Usually, the impedance control strategy is used in the robotic control research area, especially for robot–human collaboration applications [39]. This method enables robots with flexible contact force to react with human workers, avoiding collisions and accidents during the operation process. With the development of robotic techniques, some researchers found that industrial robots can also be used for polishing because of their low cost, high flexibility, and programmability for complex tasks [19,20,34,40]. Then, the position-based active impedance control technique is utilised in the robotic polishing area. To suppress the contact force fluctuations and vibrations during the polishing process, Chen proposed an integrated smart end effector mounted on a robotic arm [40]. In this way, the robotic polishing system can regulate the polishing contact pressure by cooperating with an active end-effector. In this kind of robotic compliance control, the compliance-end-effector is usually mounted at a robot arm’s end [41]. However, for the polishing process under non-laboratory conditions, the structure of this end-effector is highly complex and challenging to maintain. Also, a force sensor connects with the end-effector by cantilever structure, which may conduct vibration and be limited by the payload capacity of robots. On the other hand, some researchers try to address this problem with the underlying level of robotic control. Through analysing and simplifying the robotic rigid body model, they propose kinematic and dynamic models of the polishing robot [42,43]. With these mathematical models, they can estimate the output force of each joint and end-effector of the robot. However, the polishing system is a complex and coupled system [44]. This kind of ideal simplified mathematical model makes it challenging to match the actual circumstances during the polishing process. There are many features that are impossible to extract as a constant model, for example, the disturbance from the environment and uncertainties from the polishing devices. Hence, the potential application of robot-assisted polishing technique on ultraprecision finishing is limited by the inevitable model error of industrial robots [45]. Researchers also employed online compensation strategies on a real-time level to execute constant contact force in the micro-optic polishing process. Guo and his colleagues [46] developed a polishing control system that regulates the position of the piezo stage and linear stage with a simple PID position control to enhance the stability of the polishing control system. Regarding real-time polishing control, they also proposed a real-time closed-loop polishing contact force control strategy based on the developed polishing control system. The proposed force control strategy can achieve a fast response of 1 ms and a high position resolution of 5 nm, and the controllable polishing contact force can be kept constant within a range of 0–200 mN, which is suitable for the micro-optic polishing process. However, for a general-size workpiece in the bonnet polishing process, it is difficult for a simple position controller to adapt to sudden changes in the surface curvature of the workpiece [33]. Moreover, most researchers implement impedance control with the classical PID method, even its original model, which lacks adaptivity to the unknown external environment, such as free-form surface workpieces.

To address the above issues, in this work, we propose a novel ultra-precision polishing contact pressure control strategy based on a hybrid parallel/serial mechanism and an open architecture control system to deploy it in real-time. To our knowledge, there is little work on the active compliance control strategies of the polishing or grinding deployed on the primary mechanism, and few researchers utilise the high-frequency robust control strategy to enhance the impedance control in the dynamic force control research area. Moreover, no one has employed the active impedance control strategy on a parallel mechanism for polishing. Therefore, inspired by an event-driven approach from the computer science area, we explore a direct, robust impedance force control method to regulate the contact pressure between the workpiece and tools based on the machine’s main body. With force sensors and encoders, an open architecture and flexible polishing control system is developed, mimicking the natural manual hand polishing process with online compensation and deploying on the BP polishing machine tools. The proposed method can adjust the polishing pressure according to the desired value and pre-planned trajectories, as well as the real-time feedback from the sensors. This force control method combines high-frequency robust control and impedance control to suspend the uncertainties and nonlinearities of the compliance polishing system with little prior knowledge of the workpiece, such as the size of the surface. Moreover, this is a portable solution. It just consists of several sensors, an interface board, and universal software. The end-users can set it up on a traditional machine quickly. To make it possible to follow the pre-defined toolpath, we also developed a G-code interpreter as a bridge between the conventional machine tool and the portable polishing control system.

A series of experiments were conducted to verify the proposed system. The experimental results show that the proposed active compliance control strategy can regulate the contact force, tracking the pre-defined trajectory with good response-ability and following the desired contact pressure with high accuracy. According to the experiment results, we verified Gracia’s claim that “the bandwidth of the force control should be significantly lower than the kinematic control frequency for stability reasons” [47]. The proposed active compliance polishing control system also improves the surface quality. The average surface roughness value Sa decreases from 0.057 um in the conventional method to 0.027 um in the proposed method.

The contributions of this work can be summarised as the following three points:

The original kinematic model of the parallel mechanism involves time-consuming matrix transformation and complex calculation, limiting the online computation capacity. Therefore, in this work, we optimise the kinematic model for online computing and integrate it into the developed software.

To improve the surface quality, a polishing contact pressure control strategy is proposed. Active impedance control is the core of this force control strategy, and high-frequency robust control is combined to enhance the stability of the impedance control strategy. Overall, a robust position-based active compliance controller for a hybrid polishing mechanism is designed.

To deploy the proposed force control strategy, an open architecture control system is necessary to obtain the authority access to the velocity control loop and position control loop of the control system in real-time. Therefore, a computer-based multi-axis hybrid bonnet polishing control system with interrupt response and interrupt handling abilities is developed based on QT and C++.

The rest of the paper is organised as follows: Section 2 illustrates the simplification of the kinematic model for the parallel mechanism based on the previous research. Then, the methodology of the proposed high-frequency-impedance robust force control strategy and analysis are explained. After that, the design of the presented open architecture control system for a hybrid polishing mechanism is introduced. The fourth part is the results of a series of verification experiments, including experiment settings and result discussions. The last part is the conclusion of this work.

## 2. Methodology

### 2.1. Kinematic Analysis of the Hybrid Mechanism

The design of the polishing machine and the features of the hybrid mechanism have been explained in detail in previous works [29,30]. They analysed the hybrid mechanism theoretically and presented the kinematic model with mathematical expressions comprehensively. Although these works help us understand the fundamental characteristics and features of the proposed hybrid mechanism, they have been done from an academic view. It is necessary to be simplified to deploy on a practical motion controller. Based on the earlier research, this work presents a comparatively developer-friendly kinematic model of hybrid mechanisms by modifying the setting of basic coordinates of each part of the mechanism and unifying the geometrical presentation of each actuator and moving end-effector in the same coordinate. Some formula transformation processes have also been parameterised. Therefore, the control system can reduce the number of calculational steps of matrix transformation and make the update frequency of the kinematic control loop higher. In this way, the kinematic model can be inserted into the motion control algorithm and run online. This simplified kinematic analysis deploys the mathematical model to an executed application effectively because of fewer calculation steps and faster matrix computation in the controller, which is limited by the computation power and tight operating cycles. Then, the parameterization of the kinematic model is executed to enable further adaptive parametre optimisation, which can enhance the performance of the motion controller. An illustration of the kinematic model is shown in Figure 4. The optimised kinematic model is shown in Figure 5.

According to Figure 5a, the kinematic model of the parallel mechanism can be analysed by the geometrical method. A global coordinate frame is defined to be located at the centre of the base plate (an equilateral triangle with vertices $A_{1}$, $A_{2}$, and $A_{3}$). The parameters that can describe the mechanical characteristics can be defined in sequences, such as the length of the limb, the endpoints of the moving platform ($D_{1}$, $D_{2}$, $D_{3}$), the inclination angle, the position angle of the limb, etc.

Through geometrical analysis and parameterization, we can obtain:(1)$$\vec{OC_{i}}=\vec{OA_{i}}+q_{i}\cdot \vec{e_{li}}+\vec{B_{i}C_{i}},$$
(2)$$\vec{OD_{i}}=\vec{ON}+\vec{NP}+\vec{PD_{i}},$$
(3)$$\vec{C_{i}D_{i}}=\vec{OD_{i}}-\vec{OC_{i}},$$
(4)$${\left\|C_{i}D_{i}\right\|}^{2}=L^{2},$$
(5)$$e_{l1}=\left[\begin{array}{c} -\sin\frac{\varphi }{2}\cos\alpha \\ \cos\frac{\varphi }{2}\cos\alpha \\ -\sin\alpha \end{array}\right]{\;e}_{l2}=\left[\begin{array}{c} -\sin\frac{\varphi }{2}\cos\alpha \\ -\cos\frac{\varphi }{2}\cos\alpha \\ -\sin\alpha \end{array}\right]\;e_{l3}=\left[\begin{array}{c} \cos\alpha \\ 0\\ -\sin\alpha \end{array}\right].$$

After that,
(6)$$w^{T}=\left[\begin{array}{ccc} \begin{array}{c} x-e_{l11}q_{1}+C_{w11} \end{array} & y-e_{l12}q_{1}+C_{w12} & z-e_{l13}q_{1}+C_{w13}\\ x-e_{l21}q_{2}+C_{w21} & y-e_{l22}q_{2}+C_{w22} & z-e_{l23}q_{2}+C_{w23}\\ x-e_{l31}q_{3}+C_{w31} & y-e_{l32}q_{3}+C_{w32} & z-e_{l33}q_{3}+C_{w33} \end{array}\right],$$
(7)$$w^{T}\cdot w=L^{2},$$
where $e_{li}$ denotes the unit vector of each axis, $q_{i}$ represents the position of each axis, $\left(x,y,z\right)$ is the desired point of the end-effector, α is the angle between the actuator axis and the baseplate, and φ is the angle contained by the sides of the baseplate. $C_{wij}$ is the middle variable. By using a closure equation of geometry, the control value of the position can be computed by the following formula.

The inverse kinematic model can be expressed as:(8)$${\left(x-e_{lij}q_{i}+C_{wij}\right)}^{2}+{\left(y-e_{lij}q_{i}+C_{wij}\right)}^{2}+{\left(z-e_{lij}q_{i}+C_{wij}\right)}^{2}=L^{2},\;i=1,\;2,\;3;\;j=1,\;2,\;3,$$
where *L* denotes the length of the limb.

The illustration of the serial mechanism geometrical analysis is shown in Figure 4b. The serial mechanism comprises two rotation axes, the A-axis and the B-axis. The A-axis rotates about the vertical orientation, and the B-axis rotates about the horizon orientation. The polishing head is mounted at the end of the B-axis. However, the centre of the polishing head is constant at the cross point of these two axes. These two axes regulate the incline angles of the polishing head both in the horizon and vertical directions to implement the “precess motion” of the polishing head. Therefore, the kinematic model of the serial mechanism can be analysed using the geometrical method, Euler–Rodrigues formula, and projection principle. The target attitude of the polishing head is the vector $\vec{O^{\prime }P_{a}}$, from the original attitude $\vec{O^{\prime }P}$, $O^{\prime }$ is the centre of the polishing head, $P_{a}$ is point $\left(x^{\prime },y^{\prime },z^{\prime }\right)$.

The control target $\theta_{A},$ and $\theta_{B}$ is:(9)$$\theta_{A}=arc\tan\left|\frac{x^{\prime }\cot\theta_{B}}{\gamma \cos\delta }\right|-arc\tan\left|\frac{y^{\prime }}{z^{\prime }}\right|,$$
(10)$$\theta_{B}=arc\sin\left(\frac{x^{\prime }}{\gamma \sin\delta }\right),$$
where γ is the radius of the polishing head, δ is the angle between the original attitude $\vec{O^{\prime }P}$ and the vector $\vec{O^{\prime }Z}$.

### 2.2. Principle of High-Frequency-Impedance Robust Force Control Strategy

In this section, a robust position-based compliance control strategy is proposed. The force control law follows the impedance control. The novelty of this strategy is that we combine the high-frequency robust control with the impedance control to enhance the adaptive capacity of the designed control system. The force feedback data are acquired by the control system continuously. A high-frequency robust control law collects the force error data and outputs a force adjustment value through Lyapunov’s Second Method. This force adjustment value is regarded as an input of the impedance control law. The output of the impedance control law is a position adjustment value to indirectly control the position of the mechanism regulating the contact force. The proposed high-frequency-impedance control strategy collects force data and outputs a position adjustment value eliminating the drawbacks of the estimation of the external environment stiffness.

According to previous research works, the contact process of the interaction between a rigid body and the environment can be divided into two stages, the free space stage and the contact stage [43,48,49]. In this work, we extend this concept to the polishing process. Moreover, when discussing the physical model of the mechanism in impedance control, the polishing mechanism or the actuation mechanism can usually be analysed as a second-order mass-spring-damper system. Therefore, the parallel mechanism where the workpiece is mounted can be analysed with a second-order mass-spring-damper system. The illustration is shown in Figure 6. On the other hand, although the physical model of the polishing head can also be presented as a second-order mass-spring-damper system as it is covered by a pad that consists of rubber and elastic materials, in this work, only the dynamic system of parallel mechanism will be analysed by decoupling a mass-spring-damper system to simplify the physical model. Hence, assuming the external environment of the parallel mechanism is rigid, the workpiece that is mounted at the end-effector of the parallel mechanism is controlled by a position-based compliance control strategy to interact with the external environment.

The conventional impedance control strategy is setting appropriate parameters of the impedance control model to take control of the situation between external force and position [49]. The force orientation is simplified and only applied in one direction. It can be defined as:(11)$$f_{ex}=m\ddot{e}+b\dot{e}+ke,$$
where $f_{ex}$ represents the exerted force from the external environment, e is the tracking error of the position, $x_{d}$ is the reference target position, and x is the current position, $e=x_{d}-x$, m,b,k are the coefficients of the impedance control. In robotic application scenarios, to satisfy the requirement of steady-state error approximate to zero, researchers set the spring constant k equal to zero to eliminate the steady-state error item. Hence, the original impedance control model was modified to:(12)$$m\ddot{\varepsilon }+b\dot{\varepsilon }=f_{e}-f_{d},$$
(13)$$\varepsilon =x_{e}-x,$$
where $x_{e}$ is the position of the external environment, $f_{e}$ is the measured force feedback, and $f_{d}$ is the desired contact force. ε is position perturbation, m,b are model gains.

Assuming the stiffness of the environment is $k_{e}$, then
(14)$$f_{e}=k_{e}\left(x-x_{e}\right)=-k_{e}\varepsilon ,$$

We can get
(15)$$m\ddot{\varepsilon }+b\dot{\varepsilon }+k_{e}\varepsilon =-f_{d}.$$

However, this method needs to estimate a suitable value of external environment stiffness $k_{e}$, which makes it difficult to achieve a suitable initial value in various applications. Nevertheless, the performance of the designed impedance control law strongly depends on this estimated stiffness parametre. Therefore, in this work, we follow (12), the modified function of impedance control law, employing advanced control theory to eliminate the control error instead of the assumption of external environment stiffness $k_{e}$.

In the free space stage, the workpiece mounted on the end-effector is set to move towards the polishing head or just in contact with the polishing head. Then
(16)$$m\ddot{\varepsilon }+b\dot{\varepsilon }+k\varepsilon =-f_{d},$$
(17)$$\varepsilon =x_{e}-x.$$

If the desired force $f_{d}$ is set to zero, then kε should be equal to zero. $\ddot{\varepsilon }$ and $\dot{\varepsilon }$ also equal to zero as the ε is a constant value. The ε is considered as an adjustment position for the end-effector compensating for the force error. As a result, the end-effector will stop at the current position when $f_{d}$ is set to zero and moves forward to the polishing head with the desired force $f_{d}$ when it is non-zero.

In the contact stage, the output of the system is ε, and the input of the system is the force error $e_{f}=f_{e}-f_{d}$, we can express as:(18)$$m\ddot{\varepsilon }(t)+b\dot{\varepsilon }(t)+k\varepsilon (t)=e_{f}(t).$$

Laplace transform on equation (18) is:(19)$$\frac{E_{\left(s\right)}}{F_{\left(s\right)}}=\frac{1}{ms^{2}+bs+k}.$$

To make the control system converge, the steady state error of the contact stage should be zero, which is
(20)$$e_{ss}=\underset{s\to 0}{\lim}s\cdot \left(ms^{2}+bs+k\right)=0,$$

When the steady-state error $e_{ss}=0$, k should be equal to zero.

To avoid the involvement of external environment stiffness $k_{e}$ and enhance the performance of the modelled compliance controller, a robust impedance control strategy is employed by optimising the control input estimation. In previous research, most researchers prefer to use pure proportional-derivative (PD) control or pure impedance control to regulate the output value of the impedance control model. However, in reality, the initial value problem is crucial for these control strategies, which is strongly related to the overshoot [2]. Both control laws are difficult to handle the uncertain nonlinear dynamic systems. Therefore, a robust control strategy is employed for compliance control of the parallel mechanism. The mathematical model of the robust control strategy is as follows:(21)$$\dot{x}=f\left(x\right)+u.$$

In this work, the robust control law is used to calculate the regulation value of the force control loop.
(22)$$\dot{f}=g\left(f\right)+u,$$
where $g\left(f\right)$ can be regarded as the uncertainty of the system, u is the force control adjustment value, which is used to compute a position adjustment value by impedance control law.
(23)$$m\ddot{\varepsilon }+b\dot{\varepsilon }=e_{f},$$
(24)$$e_{f}=f_{d}-f_{e},$$
where $f_{d}$ is the desired contact force, $f_{e}$ is the measured force feedback. $e_{f}$ is the force control error.

We can obtain (25) from (21) and (23)
(25)$$\ddot{\varepsilon }=-\frac{b}{m}\dot{\varepsilon }+\frac{1}{m}e_{f}.$$

Therefore, we achieve the impedance control model in this work as follows, if $x=\dot{\varepsilon }$:(26)$$\dot{x}=-\frac{b}{m}x+\frac{1}{m}e_{f}.$$

In this model, x is the output of the control model, $e_{f}$ is the instant force error. The output x can be regarded as an adjustment velocity command along the *Z*-axis. Finally, a position compensation value can be calculated based on this adjustment velocity command and the frequency of the control cycle. In this model, we try to keep the force feedback $f_{e}$ following a given force command $f_{d}$, which means the force error $e_{f}$ should equal zero. Here, we design an input $u_{f}$ to eliminate the force error according to Lyapunov’s Second Method. As a smooth output of the designed controller is necessary for reducing the oscillation of the parallel mechanism, a high-frequency robust control strategy is utilised.

Let:(27)$$u_{f}=k_{r}e_{f}+{\dot{f}}_{d}+u_{faux},$$
(28)$$u_{faux}=\frac{p_{\left(f\right)}^{2}e_{f}}{p_{\left(f\right)}\left|e_{f}\right|+\vartheta }\;0<\vartheta <1\;p\left(f\right)>\left|g\left(f\right)\right|,$$
where $p_{\left(f\right)},\vartheta ,$ and $k_{r}$ are hyperparameters of the proposed high-frequency-impedance robust control, and $u_{faux}$ is the auxiliary control item.

Assume that there is a function $V(e_{f})$,
(29)$$V\left(e_{f}\right)=\frac{1}{2}e_{f}^{2},$$
(30)$$\dot{V}\left(e_{f}\right)=e_{f}\cdot {\dot{e}}_{f}=e_{f}\left({\dot{f}}_{d}-g\left(f\right)-u_{f}\right).$$

From (27)–(30),
(31)$$\dot{V}\left(e_{f}\right)=e_{f}\cdot {\dot{e}}_{f}\leq -ke_{f}^{2}+p_{\left(f\right)}\left|e_{f}\right|-e_{f}\frac{p_{\left(f\right)}^{2}e_{f}}{p_{\left(f\right)}\left|e_{f}\right|+\vartheta },$$
(32)$$\dot{V}\left(e_{f}\right)\leq -ke_{f}^{2}+\vartheta (\frac{p_{\left(f\right)}\left|e_{f}\right|}{p_{\left(f\right)}\left|e_{f}\right|+\vartheta }),\;0\leq \frac{p_{\left(f\right)}\left|e_{f}\right|}{p_{\left(f\right)}\left|e_{f}\right|+\vartheta }\leq 1.$$

Therefore, we can get:(33)$$\dot{V}\left(e_{f}\right)\leq -ke_{f}^{2}+\vartheta ,\;\left|e_{f}\right|\leq \sqrt{\frac{\vartheta }{k}}.$$

As 0<ϑ<1, we can know that $V\left(e_{f}\right)$ is positive definite, $\dot{V}\left(e_{f}\right)$ is negative definite. The system is asymptotically stable. If ϑ=0, the system degenerates into a sliding control system, and when ϑ≠0,
(34)$$\left|\frac{p_{\left(f\right)}^{2}e_{f}}{p_{\left(f\right)}\left|e_{f}\right|+\vartheta }\right|<p_{\left(f\right)}.$$

The high gain means that there is a large input to regulate the uncertainty, and the high frequency means this system can provide a smooth output instead of the oscillation output from the original sliding control.

In the case of a constant force command, the term ${\dot{f}}_{d}$ should be zero. In this cascaded way, we can calculate the position command adjustment value of the *Z*-axis and correct the polishing mechanism toolpath command online to guarantee a stable and ideal polishing pressure. With the decrease of ϑ, force tracking error becomes smaller, while the input of the control system will increase. Hence, a trade-off of force tracking error and oscillation should be considered by selecting suitable settings for the proposed force controller. By this method, the closed-loop force control system is globally uniformly ultimately bounded. The parallel polishing mechanism can follow a trajectory in space with a pre-defined displacement, which means that the polishing mechanism could track a desired polishing toolpath with a pre-defined polishing contact force, ignoring external disturbance, such as changes in the surface curvature of the workpiece. The control flow is shown in Figure 7.

### 2.3. Design of the Open Architecture Control System

To implement the designed force control strategy online, we designed and developed a computer-based open architecture control system for the hybrid polishing mechanism. In the designed control system, the parameterised kinematic model and presented force control strategy are deployed to the outermost control loop. In each control cycle, an adjustment vector, which consists of a position value and a velocity value, is calculated through the force control algorithm, which is based on the feedback force measurement. Then, the adjustment vector is added to the original vector from the reference command to obtain a new reference position and velocity data. After that, the new reference vector will be sent to the kinematic model to generate the actual position and velocity command for each axis of the mechanism. There is no need to build the whole control system from scratch. Our work focuses on some core components of the system, such as mathematic models and applied algorithms. Therefore, the driven components and motion controller in this system are selected following the design characteristics and purchased from the general suppliers, increasing the compatibility of the system. The detail of the designed control system is shown in Figure 8. In this work, a customised software is developed with the QT framework to integrate and schedule all these tasks using the multi-thread method. The generated position and velocity commands are used to drive the servo system through API functions of the motion controller, which are called by the customised software running in the background. The flowchart of the software is shown in Figure 9.

Typically, the machine follows the G-code commands without interruption until the process ends. In contrast, in this work, the system should respond to the interrupts because of the requirement of some online correction, for example, interrupts from the force controller. To solve this problem, a particular control flow is designed to integrate the force control loop into the position control loop.

As shown in Figure 9, during the polishing procedure, the force error is checked in each force control loop, and the output of the force control loop is regarded as the input of the position loop. The bandwidth of the position loop and force loop is different, and the frequency of the force loop is significantly lower than the position loop. The signal-slot technique from QT is responsible for dealing with external interrupts.

To enable the generality of the proposed polishing control system, G-code interpretation should be considered. Hence, a G-code auto-generator, which consists of a G-code decoder and a trajectory planner, is developed to transmit the uploaded toolpath to the position and velocity command lists of the motion control sub-system. In this auto-generator, interpolation algorithms and cubic spline trajectory planning are used to execute the given polishing trajectories. Users input hyperparameters of the high-frequency impedance force control and polishing process settings through the interfaces. Also, this software can help users operate the polishing machine with buttons. After the uploading, the original toolpath is displayed and interpreted by the G-code auto-generator.

The hardware platform of the designed control system consists of three sub-systems: a motion control sub-system, which comprises five servo drivers, five servo motors, encoders, and a motion controller; a force control sub-system, which comprises a force sensor, an amplifier, and a communication port; and the core of the control system, a computer. A high-accuracy single-axis force sensor is mounted between the workpiece fixture and the end-effector of the parallel mechanism. The data sheet of the force sensor is shown in Table 1. Therefore, a gravity compensation algorithm is implemented at the start of the polishing process. The computer sends commands to the motion controller with Peripheral Component Interconnect Express(PCIe) and acquires the states of the motion control sub-system but also communicates with the force control sub-system with a high-speed serial port to obtain the force feedback in real time. The servo driver controls the current loop and the velocity loop, while the motion controller controls each axis’s position control loop.

## 3. Experiments, Ablation Studies, and Results

A series of experiments are performed to execute an interaction process to evaluate the performance of the proposed control system. A 304 stainless steel workpiece with a saddle surface and a semirigid bonnet polishing tool are selected. The angle of inclination is 30°. Figure 10 shows the interaction process experiments. Figure 10a shows the free contact phase.

The A and B rotation axes control the axial orientation of the polishing head tool. The workpiece moves toward the polishing tool with setting parameters. The experiments were divided into two categories: static force response experiments and dynamic force response experiments. Figure 10b shows the contact phase with reference pressures, which are 2 N and 5 N, in the static force response experiments. The static experiments focus on the force control performance in the contact and adjustment processes. Hence, the parallel mechanism of the polishing machine carrying the workpiece regulates the position of the workpiece only along the *Z*-axis. Because of the geometric errors of the parallel mechanism, the position accuracy of the end-effector is about 0.81–1.25 mm through an accuracy assessment with the aid of a double ball bar [27]. In this report, the parallel mechanism was not calibrated as we would like to know if the position error or modelling uncertainty can be purely adapted by the proposed high-frequency-impedance robust force control.

Moreover, as the measurement noise will affect the performance of the designed force control sub-system, a filter is necessary during the force online control process. In this work, median average filtering is utilised to eliminate the position oscillation caused by the noise of the force sensor feedback data. The force control sub-system with a serial port continuously collects the force sensor feedback data. The system updates the force data in each 150 ms, and this means the control frequency of the designed force control sub-system is 6.7 Hz. The relatively lower frequency matches the computational cycle, avoiding the mechanism’s oscillation and smoothing the position control curve.

### 3.1. Static Force Response Experiments

Figure 11 shows the interaction process experiments with 2 N and 5 N desired contact force, including the filtered data, such as contact force data, force error, and corresponding position adjustment of the *Z*-axis. The sampling frequency is the same as the recording frequency, which is 6.67 Hz (150 ms).

We also record the raw data from the force sensor to monitor the performance of the designed active compliance control strategy directly. In Figure 12a, we notice that the proposed control system operates the parallel mechanism to track the desired contact force with multiple stages, five stages in this experiment, at the raising stage of the contact phase. Although we only use a set of initial hyperparameters for the designed control system without a fine-tuning process, there is no overshoot at the contact phase. From Figure 12b, the recording force data shows that the proposed polishing control system can control the contact force quickly and stably.

In the traditional toolpath-based polishing process, position error inherited from the mechanism, modelling uncertainty, and variable curve of the freeform surface will cause sudden changes in the contact force. The proposed control strategy deals with these sudden force changes as external disturbances. Figure 13 shows the force data of external disturbance response experiments at 2 N and 5 N, respectively. We can find that the proposed active compliance control system responds to disturbance and returns to the former state rapidly. According to the recording force data in Figure 13a,b, we found a phase where the contact force equals zero. This is a pushed-back phase because of the applied impedance control law. The parallel mechanism has been pushed back by external disturbance and returned to the former state following the given contact force command.

To evaluate the performance of the proposed control system, some index is calculated to quantify the capability of the system. The delay time in this work is the total amount of time the proposed control system takes to respond to the change of the force command, which is calculated by the number of sample points and sampling period. For example, in the experiment of force tracking response in 2 N, there are 6 sample points accounted for in the period that the system received the changing of force command until the force data changed. The rise time in this work is the time required for the force response to rise from 10% to 90% of its final value. In this experiment, it is from 0.2 N to 1.8 N. The response time is the sum of the rise time and delay time. The details of contact force static experiment results in 2 N are shown in Table 2 and Table 3.

The rise time can be regulated by the settings of parameters of the proposed control strategy, for example, b and k in the part of the impedance control law. In this work, we chose relatively weak control parameters to smooth the regulation process, as the vibration will occur if the mechanism moves too fast during polishing, affecting the workpiece surface’s quality.

### 3.2. Ablation Study

We also implemented an ablation study to measure the contribution of the proposed high-frequency-impedance control strategy to the force control of the polishing process. We ran static force response experiments where we kept the other setting of the system the same and only changed the control flow by removing the part of the robust control law. With the implementation of the ablation study, we recorded the raw force data of the pure impedance control with 2 N and 5 N, as shown in Figure 14.

Figure 14 shows that overshoots occur at each stage of the force command shifting. Also, oscillations occur during the whole contact phase. Through this ablation study, we proved that the proposed high-frequency-impedance control strategy eliminates the overshoot effectively and smooths the force control output.

### 3.3. Dynamic Force Response Experiments

With this dynamic experiment, we compare the proposed active compliance smart control system with conventional polishing through a saddle surface finishing process. In dynamic force response experiments, the workpiece interacts with the polishing head with a motion process instead of a static point in static force response experiments. Two polishing areas were selected at the edge of the surface symmetrically. The sample is shown in Figure 15.

In this procedure, the change in the size of the contact spot caused by the curve of the surface can be regarded as an external environment disturbance. The development software generates the polishing toolpath for the comparison experiment. The tool offset is set as 0.2 mm. The polishing trajectory of these two methods is the same, which is a 40 mm straight line along the *Y* axis. The force error determines the *Z*-axis trajectory of the proposed active compliance control system. Polishing parameters are shown in Table 4.

Other settings follow the previous static force response experiments. The purely toolpath-based polishing process is implemented without calibration. Therefore, the tool offset in the process is not consistent. To make it clear, we recorded the contact force data during the entire traditional polishing process, which is shown in Figure 16. We utilise the 10 µm diamond compound (Kemet International Ltd., Maidstone, UK) in the experiments instead of a liquid slurry consisting of abrasive particles.

## 4. Discussion

From Figure 16a, we can find that the contact force records relative to the geometrical characteristic of the saddle surface. As a result, in the conventional polishing process experiment, we can presume that the surface curve changes the offset of the polishing head and the contact spot size should also be changed. All these factors will affect the quality of the surface. As shown in Figure 16b, the contact force is controlled in a certain range during the polishing process through the proposed active compliance polishing method. At the beginning of the contact stage, we can notice that there is a short time period that has frequent fluctuations of contact force. With the polishing process going on, the change rate of contact force decreases because of the presented active compliance control strategy. Also, as the workpiece keeps moving in the polishing process, the external disturbance is not a constant input, and then the detected force data changes frequently instead of a periodic fluctuation profile in the static experiments. Nevertheless, the error of contact force is controlled in 1 N for almost the entire polishing process. Hence, we can say that the proposed active compliance control system can adapt the surface curve and control the contact force within a certain range without a predefined reference. It overcomes the disturbance from the external environment and adjusts the contact force along the Z-direction directly, ignoring the position error inherited from the machine.

The polishing experiments show that the proposed force control system can decrease the surface roughness effectively. The surface roughness is measured on a Zygo Nexview white light 3D interferometer in the experiments. We measured the purely toolpath-based polishing area and active compliance polishing area, respectively. The magnification of the object lens is 40. The lateral and vertical resolutions are 208.8 nm and 0.1 nm, respectively. We chose the arithmetic average surface roughness (Sa) to illustrate the quality of the surface, which is defined according to the ISO25178 standard [50]. The surface roughness is analysed with the software MX (version 6.1.0.4). A nine-order polynomial filter is used, and other settings follow the default settings. The initial and final surface roughness is shown in Figure 17. The best value of Sa of these two polishing strategies is basically equivalent to each other, which is 0.012 um. The toolpath includes the X, Y, and Z axes for purely toolpath-based experiments, while only the X and Y axes for the proposed method. The green area is polished by the conventional method, and the red area is polished by the proposed method. Sixteen serial measurement areas in each polishing area are selected, and the interval distance of each area is 2.5 mm. The details of the generated toolpath and measurement areas are shown in Figure 18. Also, the local normal direction of the contact spot in each measurement region is calculated, which obtained the tool contact local angle in each measurement area. The comparison of surface roughness is shown in Table 5 and Figure 19.

Nevertheless, the surface consistency of the proposed method is better. Regarding the measurement of surface roughness, we chose a line through the whole polishing area, and 16 measurement points were equally divided along this line. The selected lines in the two polishing areas are symmetrical. From the first value of the measurement area, which is 0.076 um and 0.74 um, in Figure 19, we can find that, at the beginning of the two polishing experiments, the average surface roughness value Sa is at the same level as the original condition. As the process went on, the Sa value decreased. In the middle of the polishing area, such as the No. 10 measurement area, it has the best quality; however, it has the highest contact force area during the toolpath-based polishing process. Then, the Sa value increases again at the end stage of both polishing processes. It is difficult to record the offset changes of the polishing tool during the polishing process, but we can observe it through contact force indirectly. It is reasonable to infer from the record of contact force and changes in Sa value that the tool offset has changed in the traditional polishing process.

From Table 5, the average Sa achieved by the proposed method is lower than the conventional method. Nevertheless, we must point out that the position error of the polishing machine should not exceed a threshold. For instance, the threshold is 1.5 mm in this work. The position error of the X and Y axes will also affect the quality, for example, the offset of the polishing trajectory in the XY plane.

## 5. Conclusions

In this research, an optimised kinematic model of the hybrid mechanism is presented to enable the kinematic model to run online. With this improved kinematic model, the control system can implement the kinematic operation faster, which is fundamental to the proposed active impedance control strategy. Moreover, through Lyapunov’s second method, a novel high-frequency-impedance robust force control strategy is proposed. The proposed active compliance control strategy enables the polishing mechanism to track a desired polishing toolpath with a pre-defined polishing contact pressure, ignoring external disturbance. This method outputs the position adjustment directly just by acquiring force feedback data instead of employing the environment stiffness estimation procedure and system model parameter modification. A computer-based open architecture control system for a hybrid polishing mechanism was designed to employ the proposed force control strategy responding to external interrupts. Customised software was developed to integrate various algorithms and sub-systems.

The performance of the designed control system was testified by a series of interaction experiments. The experiment results show that the proposed control system can adjust the contact force in real time and respond well to external disturbance. In static force response experiments, there is no overshoot at the contact phase, and the offset of the force is controlled within 0.1 N. The delay time of the proposed system is 0.6 s, which consists of the delay time of the motion control system, driven system, and mechanism components. The proposed control strategy can respond to external disturbance rapidly and stably. The push-back phenomenon can also be explored in further research for the manufacturing of brittle materials. Through the comparison experiments of the conventional toolpath method, we found that the proposed polishing strategy can control the contact force within 1 N in the whole polishing process and achieve better surface consistency, which decreases from 0.057 to 0.027 um.

The proposed control system can optimise the performance of the manufacturing process with force control by itself at the control system level, so it may be considered a self-optimisation system. Although this control system is developed for this hybrid polishing mechanism, it can also be utilised in other applications, such as milling or grinding. Also, this is a portable solution that is easy to modify and deploy on a classical machine tool, helping industries improve their manufacturing performance.

## Figures and Tables

**Figure 1 sensors-24-00421-f001:**
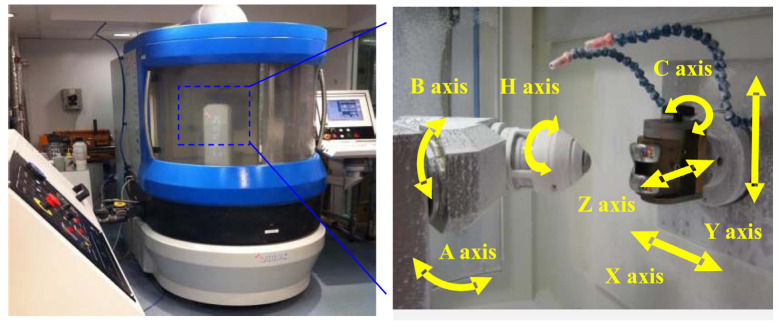
Bonnet polishing machine—Zeeko IRP series for Ultra-precision freeform polishing [26].

**Figure 2 sensors-24-00421-f002:**
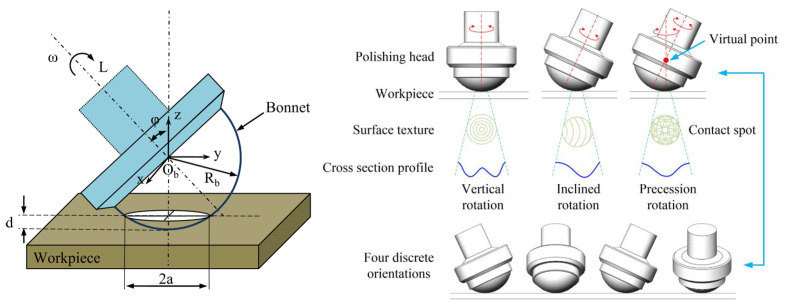
Schematic diagram of bonnet polishing and geometric parameters [26,27].

**Figure 3 sensors-24-00421-f003:**
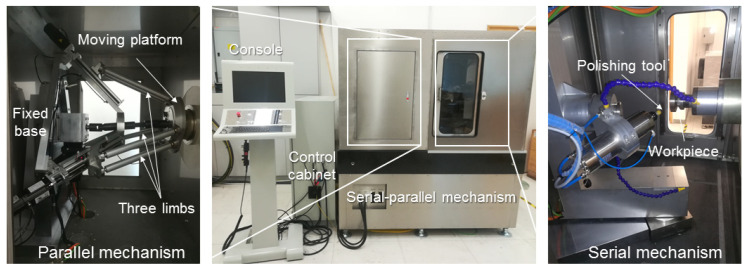
Prototype of the Hybrid Polishing Mechanism [29].

**Figure 4 sensors-24-00421-f004:**
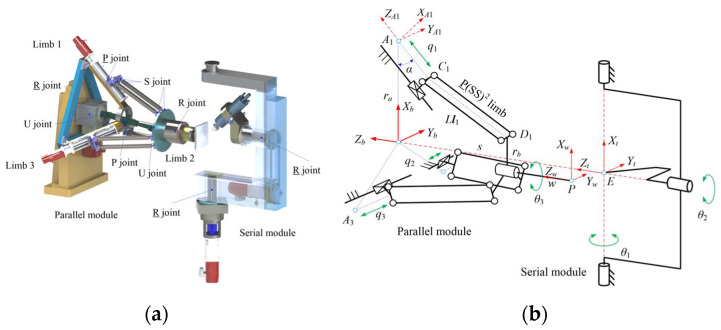
Illustration of the kinematic model of hybrid mechanism [27]: (**a**) Simulation module and (**b**) Analysis module.

**Figure 5 sensors-24-00421-f005:**
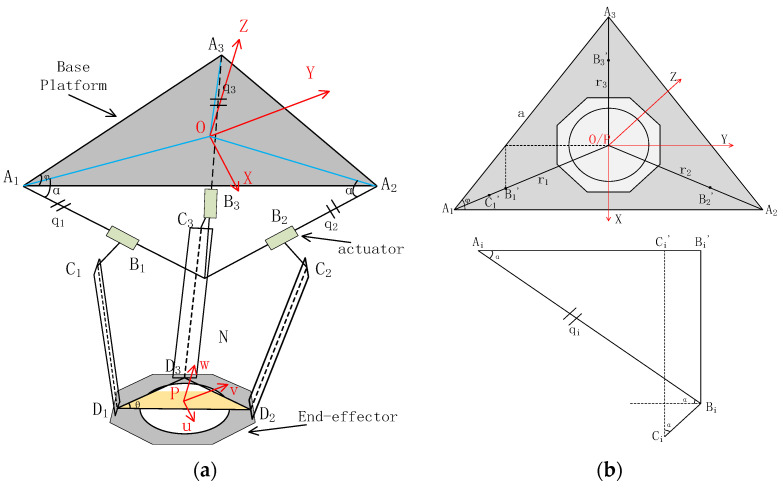
Optimised kinematic model of parallel mechanism: (**a**) Overall view of the parallel mechanism and (**b**) Simplified view of the parallel mechanism.

**Figure 6 sensors-24-00421-f006:**
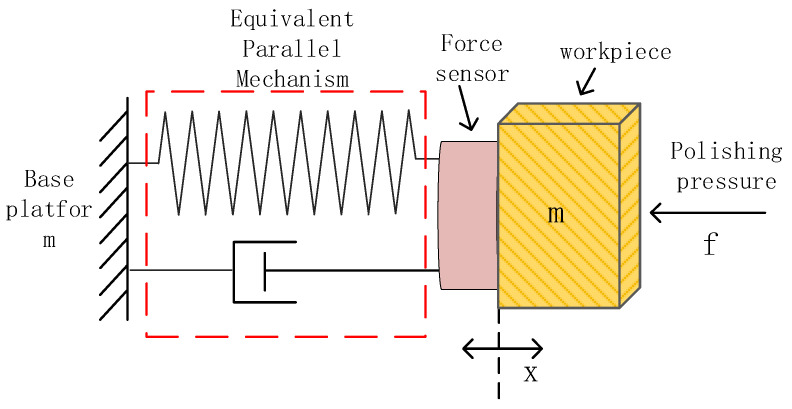
Illustration of the contact stage in compliance control during the polishing process.

**Figure 7 sensors-24-00421-f007:**
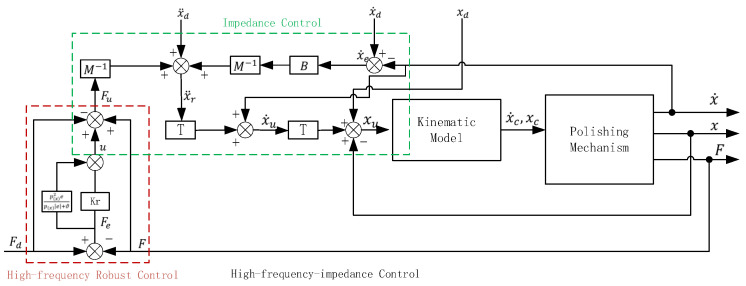
Control flow of proposed control strategy.

**Figure 8 sensors-24-00421-f008:**
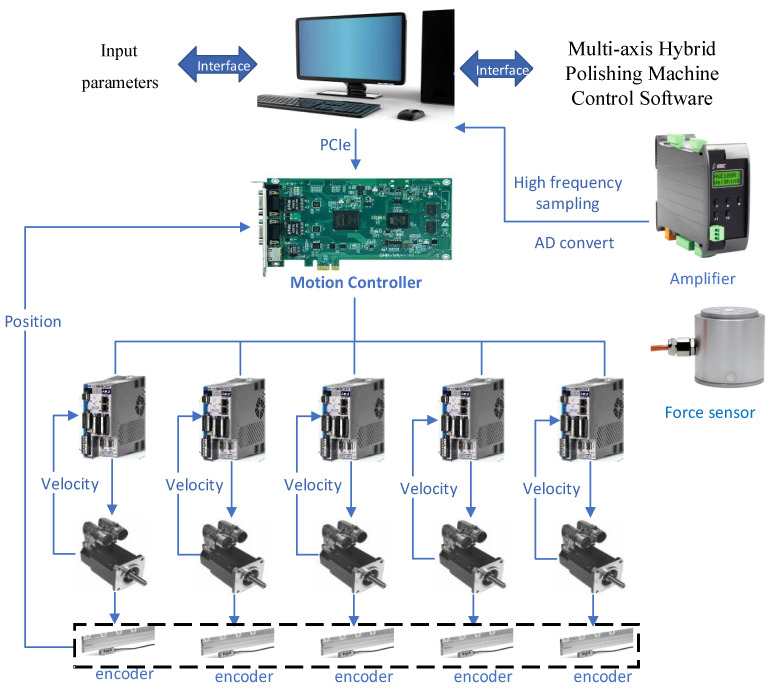
Schematic of the open architecture control system.

**Figure 9 sensors-24-00421-f009:**
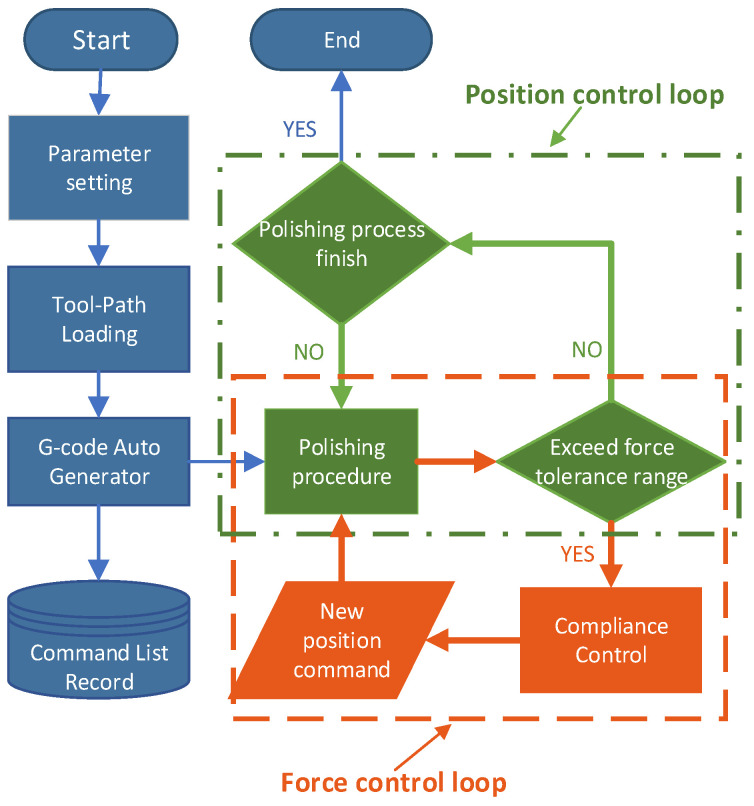
Flowchart of the designed open architecture control system.

**Figure 10 sensors-24-00421-f010:**
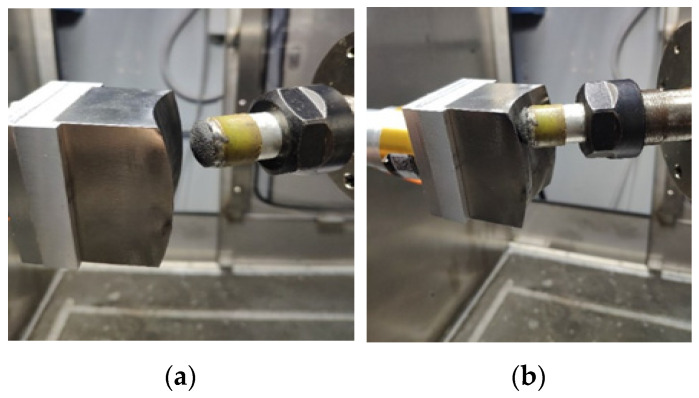
Interaction process experiments: (**a**) Free contact phase and (**b**) Contact phase with a reference pressure.

**Figure 11 sensors-24-00421-f011:**
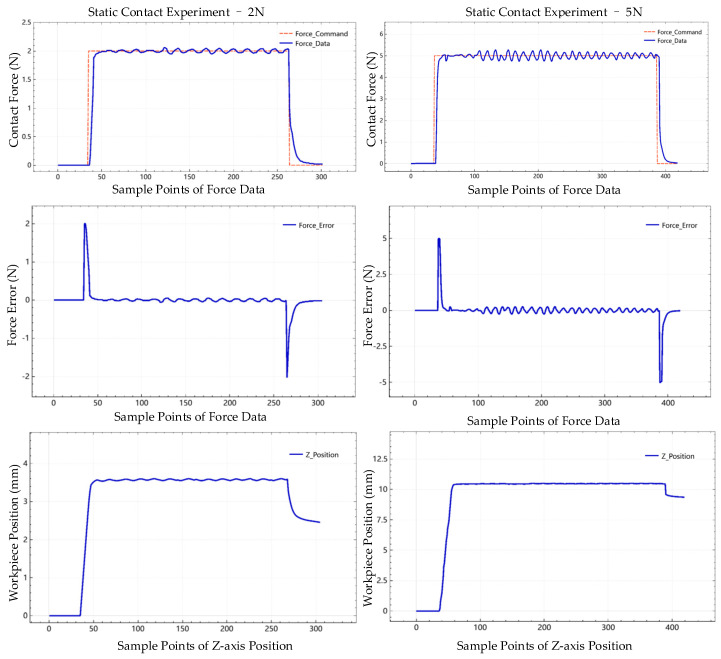
Filtered data in interaction experiments with 2 N and 5 N contact force.

**Figure 12 sensors-24-00421-f012:**
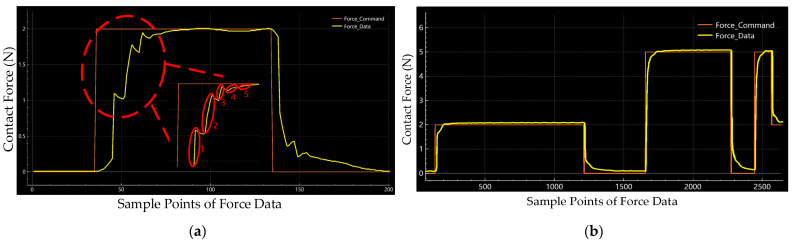
Raw data of force tracking experiments in (**a**) Real-time curve and five steps of contact phase in 2 N and (**b**) 5 N real-time curve.

**Figure 13 sensors-24-00421-f013:**
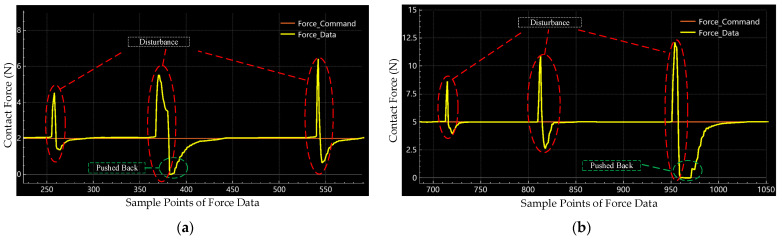
Raw data of external disturbance response experiments in (**a**) 2 N and (**b**) 5 N.

**Figure 14 sensors-24-00421-f014:**
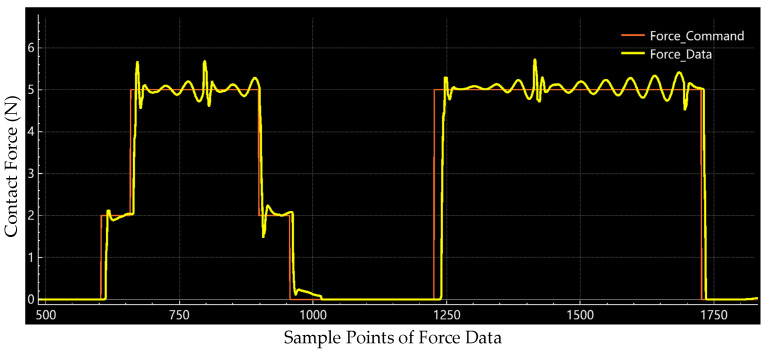
Raw data of pure impedance control at 2 N and 5 N in the ablation study.

**Figure 15 sensors-24-00421-f015:**
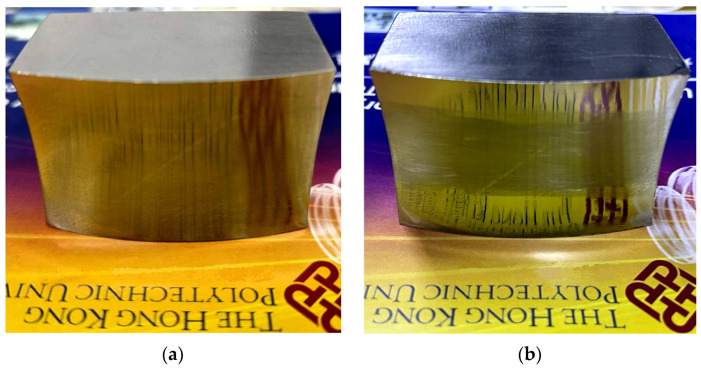
Saddle surface for polishing experiments: (**a**) Before polishing and (**b**) After polishing.

**Figure 16 sensors-24-00421-f016:**
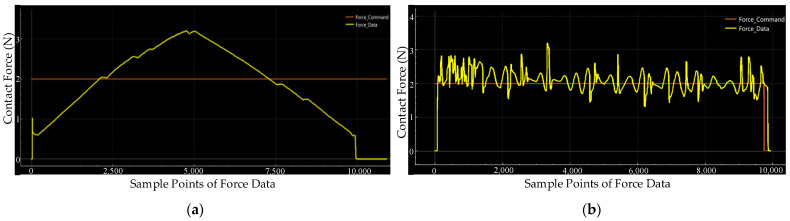
Force record of dynamic force response comparison experiments: (**a**) Force record of the conventional toolpath-based polishing process and (**b**) Force record of the proposed polishing process.

**Figure 17 sensors-24-00421-f017:**
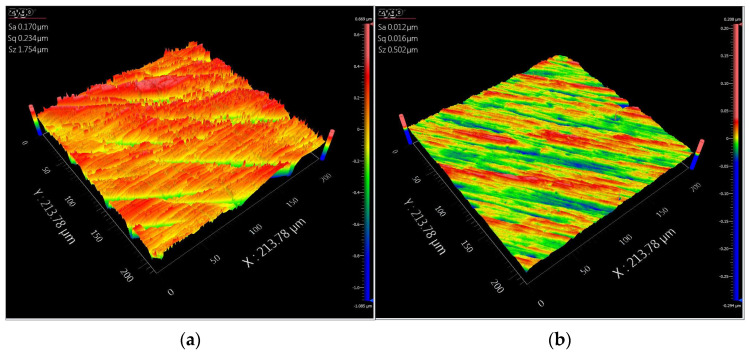
Surface quality of saddle surface: (**a**) Before polishing and (**b**) After polishing with proposed active compliance control strategy.

**Figure 18 sensors-24-00421-f018:**
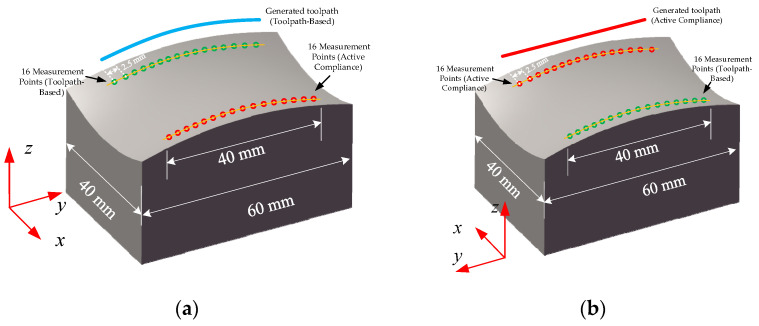
Generated toolpath and measurement areas for the comparison experiments: (**a**) Toolpath-Based polishing and (**b**) Proposed active compliance polishing.

**Figure 19 sensors-24-00421-f019:**
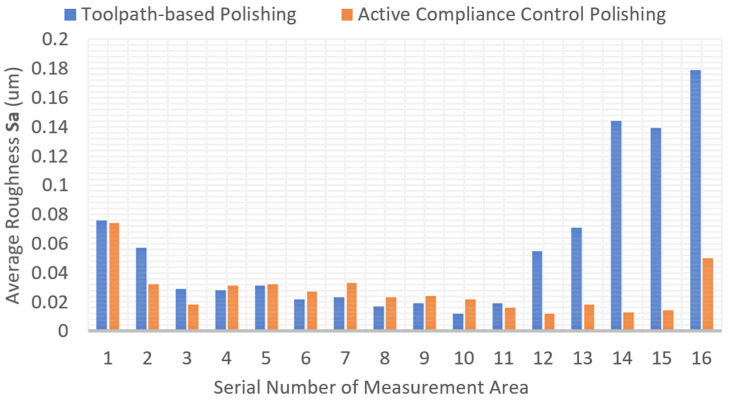
Comparison of surface quality of two polishing strategies. Sixteen serial measurement areas in each polishing area are selected, and the interval distance of each area is 2.5 mm.

**Table 1 sensors-24-00421-t001:** Properties of force sensor.

Parameters	Units	Value
Force range	KN	0–50
Sensitivity	mV/V	2.0 ± 10%
Linearity	%F.S.	0.5
Hysteresis	%F.S.	0.1
Repeatability	%F.S.	0.1
Null shift	%F.S./10 °C	0.1
Null offset	%F.S./10 °C	0.1
Response frequency	Hz	1 k
Resistance	Ω	350/700
Operating Temperature	°C	−20–80

**Table 2 sensors-24-00421-t002:** Force Tracking Response in 2 N.

Index	Time (s)
Response time	5.4
Delay time	0.6
Rise time	4.8

**Table 3 sensors-24-00421-t003:** The offset of the Force Tracking Response in 2 N.

Index	Force (N)
Target Force	2
Maximum	2.09
Minimum	1.99
Average	2.07

**Table 4 sensors-24-00421-t004:** Critical parameters for polishing comparison experiments.

Parameters	Unit	Value
Tool radius	mm	10
Tool offset	mm	0.2
Feed rate	mm/min	120
Spindle speed	rpm	2000
Spot size	mm	0.4
Trace space	mm	0.4

**Table 5 sensors-24-00421-t005:** Experiment results of average roughness parameter Sa.

Num	Sa in Conventional Toolpath-Based Strategy (um)	Sa in Proposed Compliance Strategy (um)	Tool Contact Local Angle in Measurement Region (Degree)
1	0.076	0.074	64.76
2	0.057	0.032	64.00
3	0.029	0.018	62.83
4	0.028	0.031	62.02
5	0.031	0.032	60.78
6	0.022	0.027	59.92
7	0.023	0.033	58.61
8	0.017	0.023	57.73
9	0.019	0.024	56.37
10	0.012	0.022	55.45
11	0.019	0.016	54.06
12	0.055	0.012	53.13
13	0.071	0.018	51.72
14	0.144	0.013	50.77
15	0.139	0.014	49.34
16	0.179	0.05	48.39
Average	0.057	0.027	\

## Data Availability

The data presented in this study are available on request.

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
