# Peer review of "Active Compliance Smart Control Strategy of Hybrid Mechanism for Bonnet Polishing"

_sensors, 2024, doi:10.3390/s24020421_

Round 1

Reviewer 1 Report

Comments and Suggestions for Authors

In this paper, a high-frequency-impedance robust force control strategy is proposed, and an open control system with customized software is developed to respond to external interruptions in the polishing process, and an active compliance control strategy is implemented on the hybrid mechanism. Through this method, the hybrid mechanism can automatically adapt to the external environment under a given contact pressure without estimating the environmental stiffness. The experimental results show that the proposed strategy improves the surface quality. The logic of this article is clear. However, some problems still need to be solved before receiving.

1.The quality of some pictures in the article is poor. 

2.The title number of the text is wrong ( line 198 ).

3.What is the significance of placing Figure 9 ?

4.The placement of Figure 10 is not standardized.

5.English needs to be improved, some verb passive tense errors ( e.g., line 195; is introduced ).

Comments on the Quality of English Language

Minor editing of English language required

Reviewer 2 Report

Comments and Suggestions for Authors

In this research, a high-frequency-impedance robust force control strategy is proposed. It outputs a position adjustment value directly according to a contact pressure adjustment value. An open architecture control system with customised software is developed to respond to external interrupts during the polishing procedure implementing the active compliance control strategy on a hybrid mechanism. Experimental results show that the proposed strategy improves the surface quality. Therefore, I recommend this paper to be accepted after the following remarks revised:

(1) It is recommended that the current case studies listed below, which are considered to be directly related to the subject of the study, should also be evaluated.

https://doi.org/10.1016/j.ijmachtools.2021.103827

https://doi.org/10.1016/j.jclepro.2022.132898

(2) Figure 1. is not clear enough. Suggest replacement.

(3) What are the advantages of this polishing method compared to other surface polishing methods? Such as shear thickening and polishing。

(4) 0.0575625 and 0.0274375 are the average values of the relevant data. Suggest retaining only three significant digits.

(5) The conclusion is a data presentation of all the results of the entire paper, representing the actual work done by the entire study and its practical contribution to the current research. The background section should be explained in the abstract section. Suggest reorganizing the conclusion section.

Comments on the Quality of English Language

Minor editing of English language required.

Reviewer 3 Report

Comments and Suggestions for Authors

The article "Active Compliance Smart Control Strategy of Hybrid Mechanism for Bonnet Polishing" covers the current topic of polishing spatial elements. The topic is important because this process widely used and creates a lot of difficulties.

The proposed polishing strategy based on local force measurement is somewhat reminiscent of floating machining technologies, where the processed surface is the machining datum to which the local material remove also relates. These are efficient machining techniques, but the problem is maintaining the shape of the workpiece. If the semi-finished product is unprecision, this defect is transferred to the processed element. In this type of work, it would be necessary  to focused on the parameters of surface waviness and shape defects of the surface.

Detailed comments.

Line 25 - what is the practical purpose of giving the roughness parameter values with 7 decimal places, if 3 are enough for practical purposes? Similarly in line 174.

There are many symbols in the formulas and a comprehensive list of them would be useful.

Figures 1 and 2 are not sufficient. A scheme of the polishing process should be included specifying its parameters.

Figure 9 is illegible and does not provide any useful data.

  The Figure 15 shows the workpiece, it should be presented on a uniform background without unnecessary inscriptions.

The data contained in Fig. 18 and Table 5 should be related to the drawing of the processed element with the indication of the tool contact  local angle. In addition to the roughness measurement results, they should also make out parameters for setting the measuring device, such as cut off and etc.

The work should also include a drawing of tool paths generated by the machining program projected in the workpiece.

Round 2

Reviewer 3 Report

Comments and Suggestions for Authors

In Figure 15, if there are difficulties in removing the background, these drawings could be cropped, limiting the distracting influence of background elements.